



# A long-term dataset of debris-flow and hydrometeorological observations from 1961 to 2024 at Jiangjia Ravine, China

Li Wei [1], Dongri Song [1,2], Peng Cui [1], Lijun Su [1], Gordon G. D. Zhou [1], Kaiheng Hu [1], Fangqiang Wei [3], Yong Hong [1], Guoqiang Ou [1], Jun Zhang [1], Zhicheng Kang [1,†], Xiaojun Guo [1], Wei Zhong [1], Xiaoyu Li [1], Yaonan Zhang [2,4], Hui Tang [5]

[1] Institute of Mountain Hazards and Environment, Chinese Academy of Sciences, Chengdu 610213, China

[2] National Cryosphere Desert Data Center, Lanzhou 730000, China

[3] Chongqing Institute of Green and Intelligent Technology, Chinese Academy of Sciences, Chongqing 400714, China

[4] Northwest Institute of Eco-Environment and Resources, Chinese Academy of Sciences, Lanzhou 730000, China

[5] Earth Surface Process Modelling, German Research Centre for Geosciences (GFZ), Potsdam, Germany

† deceased

Correspondence: Dongri Song (drsong@imde.ac.cn)

**Abstract**: The study of mechanisms of debris-flow formation and movement is constrained by the lack of comprehensive and long-term field monitoring data. In 1961, the Dongchuan Debris Flow Observation and Research Station (DDFORS) was established in the highly active debris-flow catchment of Jiangjia Ravine to conduct continuous field observations of debris flows. With the advancement of technology, more high-precision instruments have been employed to monitor the entire process of debris flows. This paper presents a unique and comprehensive dataset of debris flow and hydrometeorological observations collected over a 64-year period (1961-2024) at Jiangjia Ravine, China. The dataset documents 17,001 surges for a total of 278 debris-flow events, encompasses detailed measurements of kinematic parameters of debris flow, including velocity, depth, and discharge, as well as physical-mechanical parameters such as particle size distribution of debris flow, yield stress, and viscosity of debris-flow slurry. It also incorporates the induced seismic data, providing insights into the dynamic characteristics of debris flows. Furthermore, it includes continuous records of rainfall at minute intervals, soil moisture, and suspended sediment concentrations at the catchment scale. This extensive dataset provides invaluable insights into the initiation, transportation, and deposition processes of debris flows. It can be utilized to analyze flow resistance and dynamic characteristics of debris flows, to validate various computational models, to investigate the effects of debris flows on channel morphology, and evaluate the impact of climate change on sediment transport within watersheds. The dataset is publicly accessible through the National Cryosphere Desert Data Center (NCDC) (https://www.ncdc.ac.cn/) and is organized into several categories to facilitate ease of use and analysis.

## 1. Introduction

Debris flows are catastrophic mass movement phenomena that occur in mountainous regions, characterized by their multiple surges, high velocity, high-concentration of sediment, and destructive potential. These events pose a significant threat to human lives, infrastructure, and the natural environment. In-situ debris-flow observation is crucial for improving our understanding on the triggering condition (run-off generated, or landslide transition, etc.), flow dynamics, as well as for the development, calibration and validation of debris-flow models (Berti et al. 2000; Marchi et al. 2002; Cui et al. 2005; McCoy et al. 2010; Suwa et al. 2011; Coviello et al., 2015). Direct field observations yield real-time data that capture the physical attributes and behavior of debris flows, such as flow velocity, sediment concentration, and flow depth, which are not fully obtainable through remote sensing or modeling alone (LaHusen, 2005; Arattano and Marchi, 2008; Hübl and Mikoš, 2018; Hürlimann et al., 2019).

Several catchments prone to debris flows are equipped with observation systems. In the European Alps, these include Lattenbach Creek and Wartschenbach Catchment in Austria (Hübl and Moser, 2006; Hübl and Kaitna, 2010; Hübl et al., 2017), Erlenbach Torrent, Illgraben Catchment, Dorfbach Torrent, and Spreitgraben Catchment in Switzerland (Rickenmann and McArdell,2007; McArdell et al., 2007; Berger et al., 2011; Hürlimann et al., 2011, 2015; Hirschberg et al., 2021; Aaron et al.,2023; Raffaele and Jordan, 2024), Moscardo Torrent, Acquabona Creek, and Gadria Basin in Italy (Berti et al., 1999; Genevois et al., 2000; Marchi et al., 2002; Comiti et al., 2014; Blasone et al., 2015; Theule et al., 2018), and Manival Torrent and Réal Torrent in France (Navratil et al., 2013; Theule et al., 2015; Bel et al., 2015). In the European Pyrenees, observation stations are located at Rebaixader Torrent and Portainé Catchment in Spain (Abancó et al., 2014; Hürlimann et al., 2013). In Asia, stations are situated at Kamikamihorizawa Creek on Mount Yakedake in Japan (Suwa et al., 1993; Okano et al., 2012; Ikeda et al., 2023), Shenmu Village and Yusui Stream (Yin et al., 2011; Liu and Wei, 2024), and Jiangjia Ravine in the Xiaojiang River Catchment in China. Additionally, the Chalk Cliffs in Colorado Rocky Mountains in the United States are continuously monitored (Pierson ,1986; Coe et al., 2010; McCoy et al., 2011, 2012). Most debris-flow monitoring systems have been operational for less than 30 years.

Despite the extensive investment on debris-flow monitoring, the availability of publicly accessible debris-flow datasets remains limited (Lapillonne et al., 2023). Among these datasets, notable examples include those from the Moscardo Torrent, Dorfbach Torrent, Spreitgraben Catchment, Lattenbach Creek, and Illgraben Catchment. Marchi et al. (2018) compiled a dataset encompassing 809 debris-flow events across 537 basins in the mountainous regions of northeastern Italy from the mid-19th century to 2016, including debris-flow volume, year of occurrence, drainage basin area, and the geographical coordinates of the basin outlets. Marchi et al. (2021) later provided debris-flow hydrographs and rainfall data for 30 events recorded in the Moscardo Torrent between 1990 and 2019. Mitchell et al. (2022) published records of discharge, flow depth, and velocity for 11 debris flow events in the Dorfbach Torrent, one event in the Spreitgraben Catchment, and 9 events at Lattenbach Creek. McArdell and Hirschberg (2020) documented dates and bulk volumes of 75 debris-flow events at the Illgraben Catchment from 2000 to 2007, while McArdell et al. (2023) extended this with debris-flow characteristics including occurrence date and time, peak-flow depth, peak-flow velocity, total volume, and bulk density for the period 2019-2022. Hirschberg et al. (2024) contributed volumetric water content measurements, water level, and pressure data for a location along the Illgraben debris-flow channel during the 2022 debris-flow season. Fan et al. (2019) recorded the debris-flow events and their triggering rainfall that occurred from 2008 to





2017 in Longmen Mountains after the 2008 Wenchuan earthquake, parts of event were with detailed
discharge, flow depth and density. Wang et al. (2022) provided date of 186 debris flows and triggering
rainfall induced by the 2008 Wenchuan earthquake.
In addition, several published academic articles provide data sheets detailing the dynamic
characteristics of debris-flow events. For example, Theule et al. (2018) presented dynamic parameters of
debris-flow surges based on LSPIV measurements, visual feature analyses from orthorectified images,
and radar sensor data collected in 2011, 2013, 2014, and 2015. Lapillonne et al. (2023) summarized
hydraulic features such as the Froude regime, velocity, flow level, and discharge for 35 debris-flow surges
gathered between 2011 and 2020 in the Réal torrent Catchment. Comiti et al. (2014) provided the main
characteristics of three debris flows occurred in the Gadria basin from 2011 to 2013. Additionally, Marchi
et al. (2002, 2021) reported debris-flow velocity, peak discharge, and volume for 30 events between 1990
and 2019.
Overall, the majority of available datasets pertains to parameters of individual debris-flow events.
There is a notable absence of long-term datasets that include both dynamic process characteristics and
physical-mechanical parameters, which are essential for a comprehensive dynamic analysis of debris-
flow processes. Continuous field observations of debris flow at the highly active Jiangjia Ravine
catchment commenced in 1961 and continue to be conducted by the Dongchuan Debris Flow Observation
and Research Station (DDFORS) (Zhang, 1993; Cui et al., 2005; Hu et al., 2011). The station was initially
established by the Dongchuan Mining Bureau to mitigate the impacts of debris flows on river blockages
and damage to the transportation infrastructure of the copper mines. It was later taken over by the Chinese
Academy of Sciences in 1972. These observations in the station focus on the initiation, transportation,
and deposition processes of debris flows. The data collected at this observatory now spans 64 years,
including starting-ending moments, velocity, depth, discharge, bulk density, particle size distribution,
yield stress, and viscosity. In addition to these parameters, long-term measurements of rainfall, cross-
sectional elevation, and other meteorological variables such as soil moisture and suspended sediment
content have also been recorded. Furthermore, seismic, and video footage on debris flows have been
collected in recent years. This dataset records detailed flow characteristics of debris-flow surges, rather
than merely the parameters of individual debris flow events. This 64-year dataset documents 17,001
surges for a total of 278 debris-flow events, with maximum surge flow discharge and volume recorded
at 8,026 m³/s and 249,112 m³, respectively.
This paper summarizes of the debris-flow observation at Jiangjia Ravine and overviews of the core
data, including kinematic parameters, seismic data, particle size distribution, yield stress, viscosity, and
other related measurements. We begin by introducing the study area and then describe the methods used
for data collection and processing. Illustrative examples of the dataset are provided to demonstrate its
utility. Finally, we review several studies that have utilized this dataset and conclude with information
on how to access the full dataset.

## 2. Study area


### 2.1 Geology

The Jiangjia Ravine is located on the right side of the Xiaojiang River, Yunnan Province in China.
The coordinates of the ravine are between 103°05′46″-103°13′01″E and 26°13′16″-26°17′13″N. The
elevation ranges from 1088 m a.s.l. to a maximum of 3269 m a.s.l. with an area of 48.6 km² and a main
gully length of 13.9 km. It is situated within the Xiaojiang fault zone, one of the most active fault systems
on the southeastern margin of the Tibetan Plateau. The region is characterized by active neotectonic



movements and intense seismic activity. The rocks are intersected by three fault systems trending north-
south, northeast, and northwest (Wu et al., 1990). This region is characterized by intense neotectonic
activity, resulting in frequent and severe earthquakes, with magnitude IX to X events occurring almost
every century. The sandstone and slate within this area are weak, highly susceptible to weathering, and
easily to be fragmented. Approximately 80% of the exposed rocks are highly fractured and slightly
metamorphosed, serving as the primary source of material for debris flows (Zhang et al., 2023).
**2.2 Geomorphology**
The topography of the watershed is higher in the east (with the highest elevation of 3269 m) and
lower in the west (with the lowest elevation of 1042 m). The shape of the watershed is wider in the east
(about 7 km) and narrower in the west (about 2 km). The watershed is characterized by high mountains
and steep slopes, with a relative elevation difference of approximately 500 meters between the
ridges and valleys (Fig.1a). Areas with slopes exceeding 25° and 35° make up 55% and 16% of the
total watershed area, respectively (Cui et al., 2005). The Jiangjia Ravine contains more than 200
tributary gullies, including 154 incised gullies and 46 gullies. The main channel is divided into three
sections, each with distinct morphological characteristics: (1) the erosion area, also known as the debris-
flow triggering zone, is 6.5 km long with an average slope of 11°; (2) the transportation area, 2.6 km long
with an average slope of 4.9°, and (3) the deposition area, 4.8 km long, has an average slope of 3.7°.
With the siltation of sediment in the main channel, the deposition area keeps increasing, while the
transportation area keeps decreasing.
**2.3 Climate**
The climate of Jiangjia Ravine is characterized by distinct dry and wet seasons, along with a
pronounced vertical climatic zonation (Chen, 1985). It can be divided into three climatic zones: (1) the
subtropical dry-hot valley region, located at elevations below 1600 m a.s.l.; (2) the subtropical and warm
temperate subhumid region, situated between 1600 and 2200 m a.s.l.; and (3) the temperate humid
mountain region, found at elevations above 2200 m a.s.l. (Guo et al., 2012).The rainy season extends
from May to October, during which more than 85% of the annual rainfall occurs, while the dry season
lasts from November to April, receiving less than 15% of the yearly total (Scott and Wang, 2003). Heavy
rains and thunderstorms are frequent in the rainy season, contributing to more than half of the annual
precipitation. The heaviest rainfall typically occurs between 2500 m and 3000 m a.s.l. Annual
precipitation in the watershed increases with elevation, while evaporation decreases with rising elevation
(Chen et al., 2011).
**2.4 Soil and vegetation**
The underlying soils in the region include torrid red soil, red soil, brown soil, yellow soil, and
mountainous brown soil, corresponding to different zones. Croplands are mainly distributed on gentler
hillslopes (<25°) near the divides and on the alluvial fan. Some steeper hillslopes are covered by sparse
shrubs or grass, while the others are barren for frequent failures (Yang et al., 2022).
Due to the weak lithology, crisscrossing faults, steep terrain, and sparse vegetation in the watershed,
landslide, and collapse activities are intense, covering an area that accounts for 61% of the total watershed,
storing 1.23 billion cubic meters of loose solid materials (Fig.1b). Weathering-derived debris accounts
for 70-80% of the loose material sources, while loose sedimentary deposits constitute 20-30% of the total
(Li and Wu, 1981). During the rainy season, debris flows frequently occur, with each event consisting of
tens to hundreds of surges (Fig. 2a). These surges, known as surge flows, are characterized by a distinct
head, body, and tail. In contrast, continuous flows lack these well-defined phases (Kang et al., 1990).
The ravine, now known as the "debris-flow museum" in China, offers optimal research conditions. It is



the site where China's earliest prototype observation and research on debris flows were initiated,
primarily to address the recurring river blockages caused by debris flows (up to 10 blockages in the year
1961) (Wu et al., 1990) (Fig.2b). Initially, basic observation devices were installed in 1961, followed by
the early-stage infrasound monitoring of debris flows (Zhang et al.,2004) .Over the past 64 years, the site
has developed into a comprehensive field laboratory for automated observation, experimentation, and
instrument validation (Fig. 1c). More information of DDFORS and its debris-flow monitoring can be
found in Cui et al. (2005) and Kang et al. (2004).

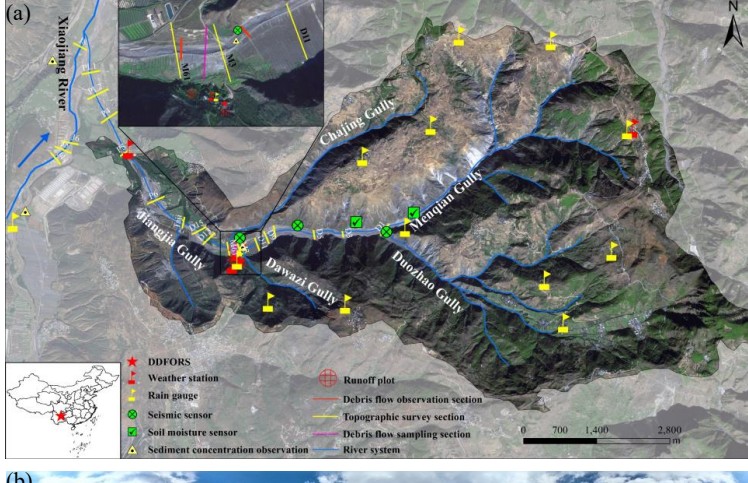


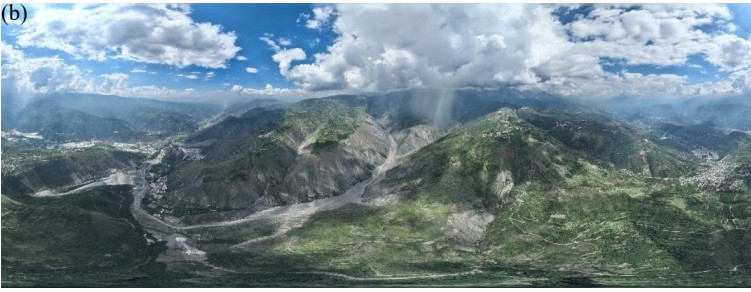


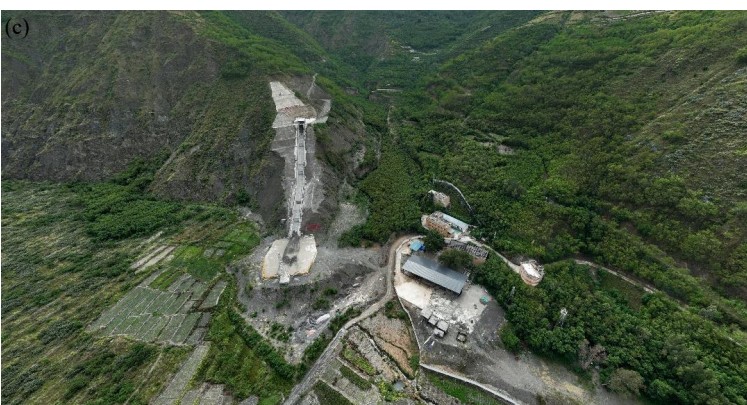


**Figure 1**. Photos of observation systems, Jiangjia Ravine and station: (a)The observation systems at





Jiangjia Ravine, satellite image obtained from https://data.cresda.cn/#/2dMap, last access: 3 March,

2025. (b) aerial view of Jiangjia Ravine, with localized heavy rainfall captured in the higher elevation

(The terrain is deformed). (b), aerial view of Dongchuan Debris Flow Observation and Research

Station (DDFORS).

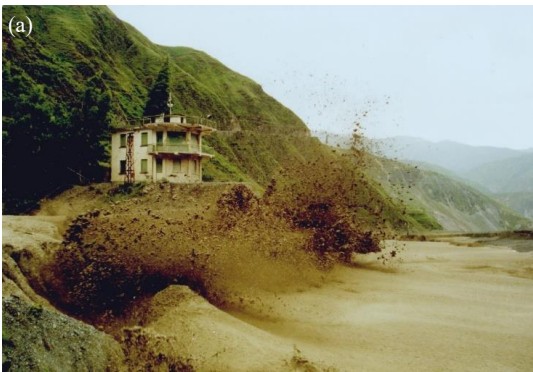

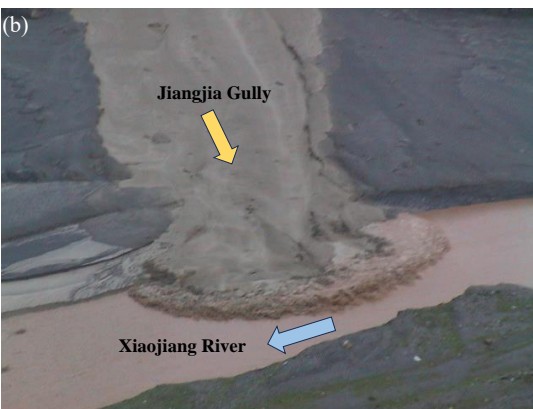

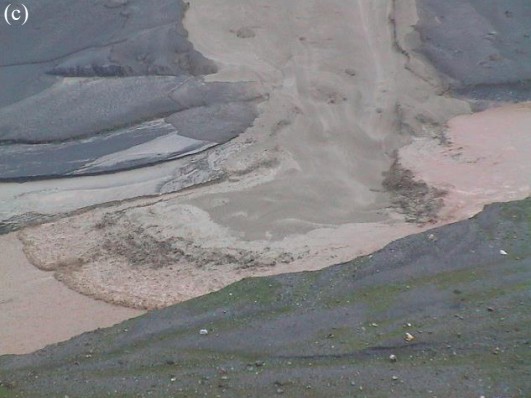

**Figure 2**. Debris flows at Jiangjia Ravine: (a) debris flow erode the channel bank (Photos by Shunli

Chen) and (b), (c) confluence of debris flows into the Xiaojiang River.



## 3. Data

The dataset comprises multiple parameters, including debris-flow kinematic data, rheological, particle size distribution data, seismic data, debris-flow video, cross-section elevation data; meteorological, rainfall, soil moisture, and temperature data; sediment concentration and runoff data from plots; suspended sediment data. The dataset comprises, but not limited to, the data from three published paper-based datasets (in Chinese), containing debris-flow observation records from 1961 to 2000 (Zhang and Xiong, 1997; Kang et al., 2006; Kang et al., 2007). Figure 1a indicates the locations of sampling or monitoring. The specific coordinates of the monitoring location can be found in the dataset. Figure 3 summarizes these datasets along with their respective acquisition periods. The majority of the data were collected at Jiangjia Ravine, while additional sediment flux, rainfall, and meteorological data were also acquired at Xiaojiang River Catchment. The instruments and measurement methods employed are detailed in Table 1.

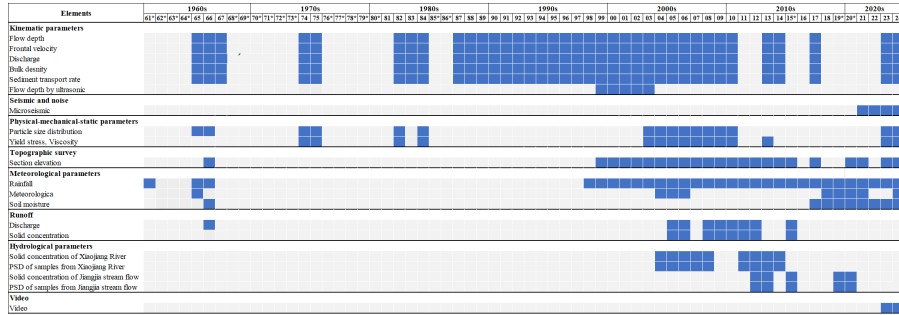

Note: Acquisition period for each parameter is shown in blue. * Denotes debris flows occurred without observational data.

**Figure 3.** An overview of the dataset.

**Table 1** Observation parameters and sensor configurations.

| Types | Elements | Parameter | Instrument | Manufacturer | Frequency Sampling interval | Accuracy | Resolution |
|---|---|---|---|---|---|---|---|
| Debris flow | Kinematic parameters | Flow depth | Determined referring to cross section marks and debris-flow level | | | | |
| | | Flow depth by ultrasonic level meter | Ultrasonic sensor | Siemens Milltronics Process Instruments Inc., Germany | 10 Hz | ±0.25% of the range or 6 mm, whichever is greater | 0.1% of the range or 2 mm, whichever is greater |
| | | Surface width | Determined referring to channel width | | | | |
| | | Frontal velocity | Determined by measure distance divided by measure time | | | | |
| | Seismic | Microseismic | Seismic sensor | Di GOS Potsdam GmbH, Germany | 100 Hz | | |
| | Static parameters | Particle size distribution | Particles larger than 0.25 mm analyzed by sieve analysis method. Particles smaller than 0.25 | Malvern Panalytical Ltd, UK | | | |



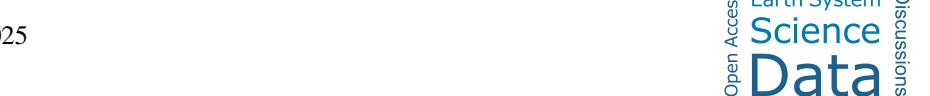
| | | | | | | | |
|---|---|---|---|---|---|---|---|
| | | | mm analyzed by pycnometer method before 2004. Marvin Laser Particle Size Analyzer (model MS2000) utilized from 2004. | | | | |
| | | Yield stress, Viscosity | Rheometer, concentric cylinder system, outer cylinder diameter 43.4 mm, inner cylinder diameter 31.44 mm. | Thermo Electron, Gmbh, Germany | | | |
| | Video | Video footage | 8 Megapixel 1/1.8 CMOS Intelligent Zoom Bullet Network Camera | Hangzhou Hikvision Digital Technology Co., Ltd, China | | | Main Stream specifications: frame rate 25 fps Resolution 3840 x 2160, at 50 Hz |
| | Topography | Section elevation | Level gauge, Real-Time Kinematic | Sichuan Leituo Technology Development Co., Ltd., China | | Horizontal direction: $\pm 8mm+1\times10^{-6}D$; Vertical direction: $\pm 15mm+1\times10^{-6}D$ | |
| Meteorological parameters | Meteorological parameters | Rainfall | Siphon rain gauge (before 2006), Tipping-bucket rain gauge (2006-2023), Piezoelectric rain gauge (2024) | Shanghai Meteorological Instruments Factory Co., Ltd., China (before 2024); Insentek technology co. ltd, China (2024) | 1 min | $\pm0.05$ (before 2006); $\pm0.04$ (2006-2023); $\pm4\%$ (2024) | 0.1 mm |
| | | Weather station | Cumulus automatic weather station | ELE International Centre of Excellence Bedfordshire, UK | 1 h | | |
| | | | Weather station | Insentek technology co. ltd, China | 1 h | Temperature: $\pm0.15$; Humidity: $\pm3\%$; Wind speed: $\pm2\%$; Wind direction: $\pm2°$; Atmospheric pressure: $\pm2°$; Solar radiation: $\pm5\%$ | Temperature: 0.02℃; Humidity: 0.05%RH; Wind speed: 0.01m/s; Wind direction: 0.1°; Atmospheric pressure: 0.1 hPa; Solar radiation: $1W/m^2$ |
| Soil moisture | Soil moisture | Soil moisture | Moisture sensor | Dalian Zheqin Technology Co., Ltd., China | 1 Hz | $\pm2\%$ within the range of 0-53%; $\pm4\%$ within the range of 53-100% | |
| Runoff plot | Runoff | Runoff volume | Measured manually | | | | |
| | | Solid concentration | Analyzed using oven drying method | | | | |
| Water sample | Hydrological parameters | Solid concentration | Analyzed using the oven drying method | | | | |
| | | Particle size distribution of | Determined by Marvin Laser | Malvern Panalytical | | | |

| | | suspended sediment | Particle Size Analyzer (model MS2000) | Ltd, UK | | | |
|---|---|---|---|---|---|---|---|

## 3.1 Debris flow

### 3.1.1 Kinematic parameters

**Measurement**

The debris-flow kinematic parameters were monitored in a straight channel adjacent to the DDFORS (Fig.4). The location of two debris-flow observation sections is shown in Fig. 1a. Observation distances ranged from 50 m to 200 m. Flow time between the two sections was measured using a second chronograph. Depth, defined as the surge front height, was determined by referencing cross-sectional marks on the channel banks, representing the vertical distance between the top of the debris-flow surge and the channel surface (Fig. 4c). Surface width of debris flow was determined by referencing of channel width. Velocity was calculated by dividing the observation distance by the flow time for each surge as it passed through the cross sections. Bulk density of each surge was determined either through sampling weighting or estimation.

During 1999-2001, the flow depth of specific debris-flow surges was simultaneously measured using an Airanger SPL ultrasonic level meter installed directly above the channel with a sampling frequency of 10 Hz (Fig. 4b). The measurement range extends from 0.3 m to 60 m. The depth value is calculated as the difference between the installation height of the ultrasonic sensors, set at 10 m, and the measured values.

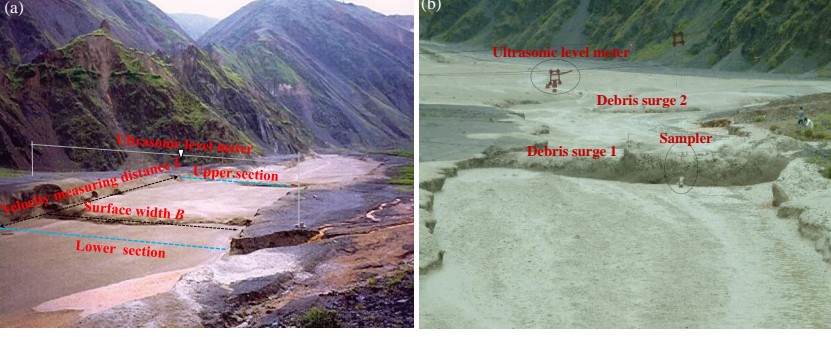

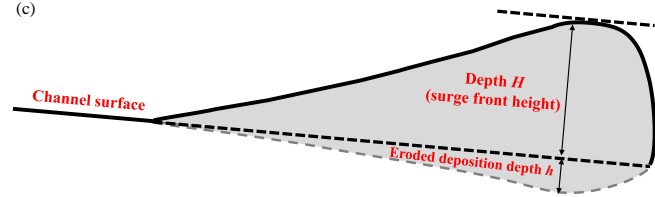

**Figure 4.** Field observation of debris-flow kinematics and sampling:(a) (b) surface width, depth and velocity measurement. (c)illustration of flow depth.

**Data processing**

The discharge, volume, sediment concentration, sediment volume, and sediment transport rate of debris-flow surge were determined based on the observational parameters.

The surge discharge $Q$ (m³/s) was calculated as following:



$$Q = V \times B \times H \tag{1}$$

where $V$ is velocity (m/s), $B$ is surface width (m), and $H$ is surge depth (m).

Surge flow volume $W_c$ (m³) was calculated as following:

$$W_c = Q \times \frac{T}{2} \quad \text{for surge flow} \tag{2}$$

$$W_c = Q \times T \quad \text{for continuative flow} \tag{3}$$

where $T$ is the record time (duration) of debris-flow surge (s), which is the time for surge front minus the time for surge rear.

Sediment concentration $S$ (kg/m³) and volume concentration of debris flow $C_v$ are obtained as following:

$$S = r_s \times C_v \tag{4}$$

$$C_v = \frac{\gamma_c - \gamma}{\gamma_s - \gamma} \tag{5}$$

where $\gamma$, $\gamma_c$, $\gamma_s$ are unit weight of water, debris flow, and sediment (grain density) respectively; $\gamma_s$ is taken as 2650 (kg/m³).

Sediment volume $W_s$ (m³) and sediment transport rate $Q_c$ (t/s) are calculated as following:

$$W_s = W_c \times C_v \tag{6}$$

$$Q_c = Q \times S/1000 \tag{7}$$

Example of kinematic parameters of debris-flow surges occurred on 7th June, 2013 are presented in Table 2. In addition, statistics of debris-flow surge parameters such as total sediment transport volume and total runoff volume are also provided.

**Table 2** Example of kinematic parameters of debris-flow surges occurred on 7th June, 2013.

| No. | Type | Time for surge front $T_1$ /(h:m:s) | Time for rear of surge $T_2$ /(h:m:s) | Duration $T$/s | Surface width $B$/m | Surge front height $H$/m | Velocity measuring distance $L$/m | Time for velocity measurement $t$/s | Front velocity $V$/(m/s) | Discharge $Q$/(m³/s) | Unit weight $\gamma_c$/(t/m³) | Sediment transport rate $Q_c$/(t/s) | Runoff Volume $W_c$/(m³) |
|---|---|---|---|---|---|---|---|---|---|---|---|---|---|
| 1 | Surge flow | 03:45:27 | 03:45:37 | 10 | 30.0 | 0.500 | 200 | 42.70 | 4.68 | 70.2 | 2.2000 | 135.24 | 351 |
| 2 | Surge flow | 03:46:15 | 03:46:27 | 12 | 50.0 | 1.000 | 200 | 29.65 | 6.75 | 337.5 | 2.2300 | 666.29 | 2025 |
| 3 | Surge flow | 03:47:26 | 03:47:36 | 10 | 50.0 | 0.700 | 200 | 34.78 | 5.75 | 201.2 | 2.2000 | 387.61 | 1006 |
| 4 | Surge flow | 03:48:40 | 03:48:52 | 12 | 50.0 | 0.500 | 200 | 33.33 | 6.00 | 150.0 | 2.2000 | 288.98 | 900 |
| 5 | Surge flow | 03:51:29 | 03:51:42 | 13 | 50.0 | 0.600 | 200 | 40.03 | 5.00 | 150.0 | 2.2000 | 288.98 | 975 |
| 6 | Surge flow | 03:54:25 | 03:54:35 | 10 | 50.0 | 0.700 | 200 | 36.83 | 5.43 | 190.0 | 2.2200 | 372.08 | 950 |
| 7 | Surge flow | 03:55:02 | 03:55:13 | 11 | 50.0 | 0.800 | 200 | 31.44 | 6.36 | 254.4 | 2.2000 | 490.10 | 1399 |
| 8 | Surge flow | 03:56:13 | 03:56:27 | 14 | 25.0 | 0.500 | 200 | 41.39 | 4.83 | 60.4 | 2.2000 | 116.36 | 423 |
| 9 | Surge flow | 03:56:40 | 03:56:51 | 11 | 30.0 | 0.500 | 200 | 46.10 | 4.34 | 65.1 | 2.2000 | 125.42 | 358 |
| 10 | Surge flow | 03:57:28 | 03:57:38 | 10 | 30.0 | 0.500 | 200 | 44.93 | 4.45 | 66.8 | 2.2000 | 128.69 | 334 |
| 11 | Surge flow | 03:58:41 | 03:58:52 | 11 | 50.0 | 1.000 | 200 | 35.63 | 5.61 | 280.5 | 2.2200 | 549.30 | 1543 |
| 12 | Surge flow | 03:59:44 | 03:59:55 | 11 | 30.0 | 0.600 | 200 | 50.67 | 3.95 | 71.1 | 2.2000 | 136.97 | 391 |





| 13 | Surge flow | 04:00:13 | 04:00:26 | 13 | 50.0 | 0.600 | 200 | 37.98 | 5.27 | 158.1 | 2.2000 | 304.58 | 1028 |
| 14 | Surge flow | 04:01:40 | 04:01:57 | 17 | 50.0 | 1.000 | 200 | 28.00 | 7.14 | 357.0 | 2.2500 | 717.11 | 3034 |
| 15 | Continuative flow | 04:02:31 | 04:04:50 | 139 | 8.0 | 0.600 | 200 | 43.56 | 4.59 | 22.0 | 2.1500 | 40.63 | 3058 |
| 16 | Continuative flow | 04:04:50 | 04:10:00 | 310 | 5.0 | 0.600 | 200 | 33.63 | 5.95 | 17.9 | 2.1500 | 33.06 | 5549 |
| 17 | Continuative flow | 04:10:00 | 04:13:28 | 208 | 6.0 | 0.500 | 200 | 28.91 | 6.92 | 20.8 | 2.1500 | 38.42 | 4326 |
| 18 | Continuative flow | 04:13:28 | 04:13:35 | 7 | 8.0 | 0.600 | 200 | 33.48 | 5.97 | 28.7 | 2.1500 | 53.01 | 100 |
| 19 | Continuative flow | 05:00:00 | 05:32:00 | 1920 | 4.0 | 0.500 | 200 | 47.38 | 4.22 | 8.4 | 2.1500 | 15.51 | 16128 |
| 20 | Continuative flow | 05:36:14 | 05:36:27 | 13 | 10.0 | 0.500 | 200 | 43.54 | 4.59 | 22.9 | 2.1500 | 42.30 | 149 |

Note: Time are provided in China Standard Time (CST, UTC+8).

**Example of results**

The kinematic characteristics of debris-flow surges provide critical information for understanding their behavior and impact. Between 1965 and 2024, a total of 278 debris-flow events were recorded, including 14,887 surge flows and 2,114 continuative flows. These events predominantly occur during the monsoon season, from May to September, with the highest frequency observed in June (55 events), July (113 events), and August (98 events).

Cumulative distribution functions of depth-flow depth, discharge, and volume for surge flows and continuative flows were depicted in Fig.5. Maximum flow depth is 5.5 for surge flow and 2.7 for continuative flow (Fig.5a; quantiles at 25%, 50%, and 75% of 0.29 m, 0.39 m, and 0.49 m for surge flows, respectively; quantiles at 25%, 50%, and 75% of 0.50 m, 0.80 m, and 1.2 m for continuative flows, respectively). The discharge of surge flows ranges from 0.5 to 8,026 m³/s, while that range from 0.3 to 1,800 m³/s for continuative flows (Fig.5b; quantiles at 25%, 50%, and 75% of 67.7 m³, 7,063 m³/s, and 1,0594 m³ /s for surge flows, respectively; quantiles at 25%, 50%, and 75% of 4.9 m³, 13.3 m³/s, and 36.3 m³/s for continuative flows, respectively). The volume of surge flows ranges from 2 to 249,112 m³, while that range from 3.4 to 825,600 m³ for continuative flows (Fig.5c; quantiles at 25%, 50%, and 75% of 3,344 m³, 6,673 m³, and 10,009 m³ for surge flows, respectively; quantiles at 25%, 50%, and 75% of 522 m³, 1,043 m³, and 1,564 m³ for continuative flows, respectively).

Figures 6 illustrates examples of measured debris-flow depth, velocity, discharge, and runoff volume of events that occurred on July 16th, 1999. The flow depth measured by ultrasonic level meter, velocity, discharge, and volume exhibit strong fluctuations, with the maximum flow depth and velocity reaching up to 5.4 m and 10.6 m/s, respectively. The volume of debris flow ranged from 403 m³ to 74,615 m³. The maximum values represent the volume of the continuative flow, which has a long duration and typically occurs during the final period of the event.

Figure 7 presents the measured annual sediment volume from 1966 to 2024. During this period, debris flows transported 43.3 million m³ of sediment. Sediment mainly deposited in the main channel, while a portion was transported to the Xiaojiang River. The most substantial debris flows occurred in 1991, transporting 6.69 million m³ of sediment. Since 2005, the volume of sediment transported has exhibited a decreasing trend.



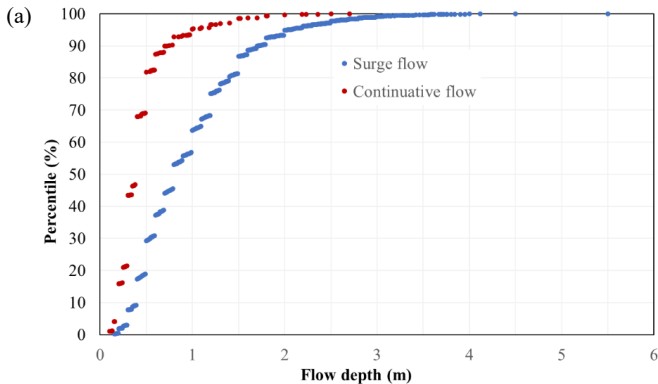


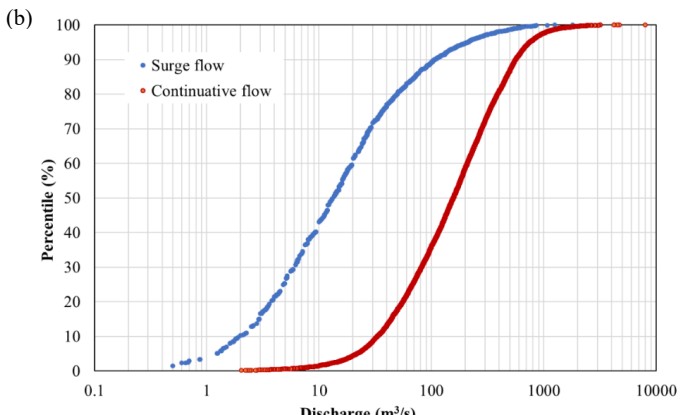


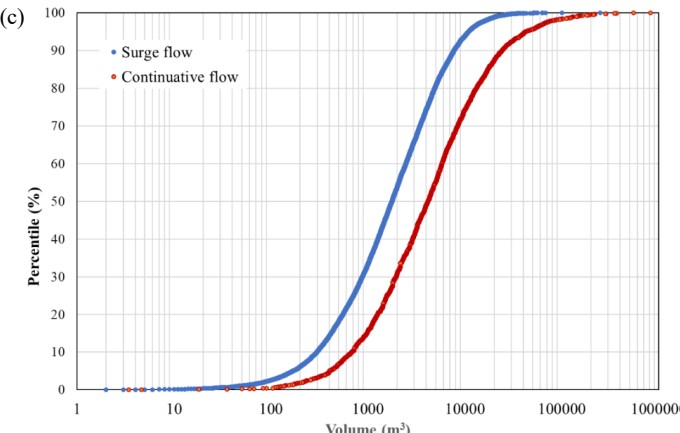


**Figure 5.** Cumulative density functions of (a) flow depth, (b) discharge, and (c) volume.





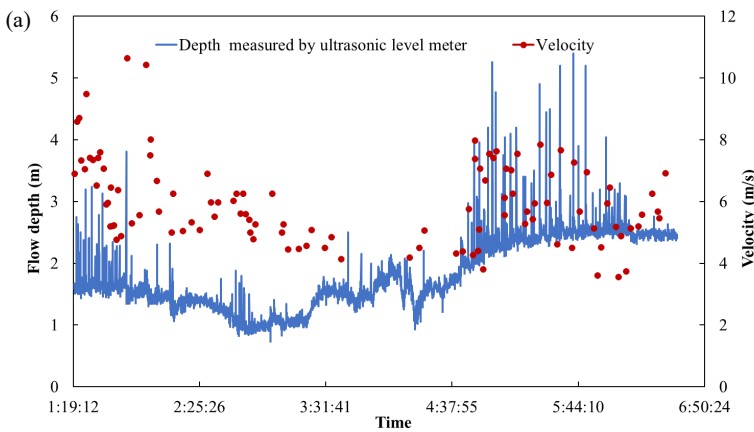


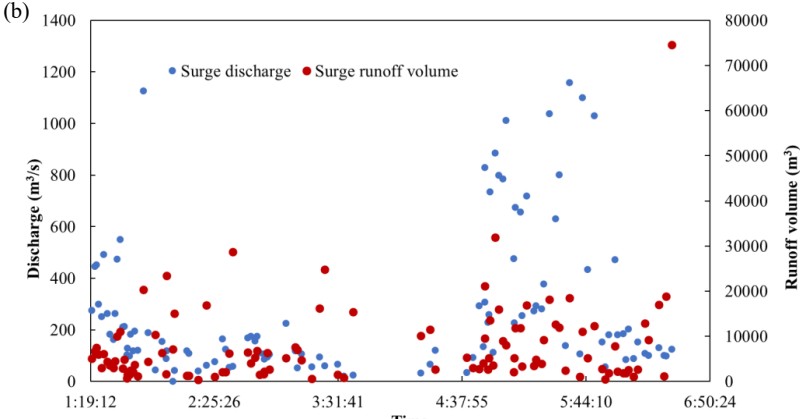


**Figure 6.** (a) Variation of flow depth, velocity and (b) discharge, volume of debris-flow surges occurred on July 16th, 1999.


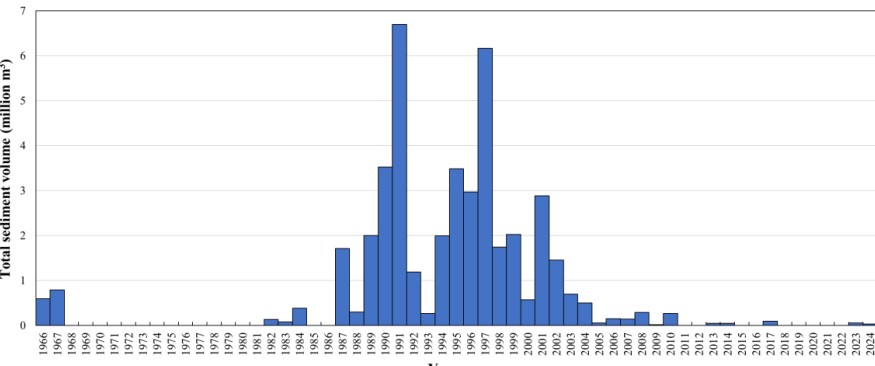


**Figure 7.** Variation of sediment transported by debris flow recorded from 1966 to 2024.




### 3.1.2 Seismic data

**Measurement**

The ground seismic response induced by debris flows was monitored by DATA-CUBE 3-channel seismic recorder. The sensors contain three geophones for different directions and were installed in three locations with average distance of 1.1 km along the main channel within the transportation area (Fig.1a, Fig.8a). The recorded seismic data was stored in the data logger of the sensors, and download manually after the debris-flow events.

**Example of results**

Figure 8b illustrates the time-domain seismic signal and time-frequency characteristics of the debris-flow event that occurred on July 28th, 2024. The event began at approximately 11:24 (UTC+8) and ended at 12:04 (UTC+8), lasting a total of 40 minutes.

From the time-domain waveforms and time-frequency spectra of seismic signals in the east-west (E-W), north-south (N-S), and vertical (Z) directions, the initiation and cessation of the debris flow are clearly identifiable. A distinct increase in signal amplitude is observed starting around 11:25, peaking at approximately 12:00, which corresponds to the main surge of the debris flow. After this peak, a significant attenuation of the seismic signal is noted post 12:00, indicating a decrease in flow intensity and the eventual cessation of the debris flow.

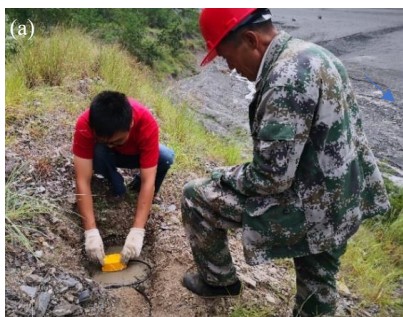

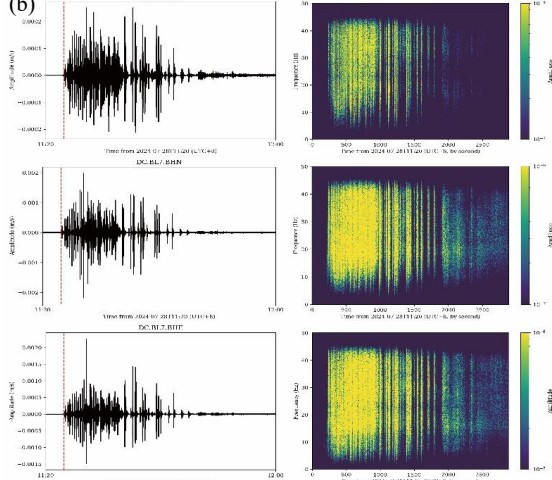

**Figure 8.** (a) Installation of seismic sensor and (b) the time-domain seismic signal and time-frequency characteristic curves of the debris-flow event on July 28th, 2024.

### 3.1.3 Particle size, yield stress, and viscosity

**Measurement**

Debris-flow samples are collected by suspending cable sampler or manual sampling. The cable sampling is collected from the moving surges with a bucket that has a diameter of 190 mm and a volume of 0.012 m³ (Fig.9). Manual sampling collects the debris-flow deposition near the channel bank by using a circular or square bucket with a volume of 0.0155 m³ or 0.009 m³, respectively.

The sample is air-dried and sieved through a steel mesh for particle size analysis. For sediments larger than 0.25 mm, the particle size is determined using the sieve analysis method. For particles smaller than 0.25 mm, the pycnometer method was employed for particle size distribution analysis before 2004. Since 2004, the Marvin Laser Particle Size Analyzer (model MS2000) has been used.

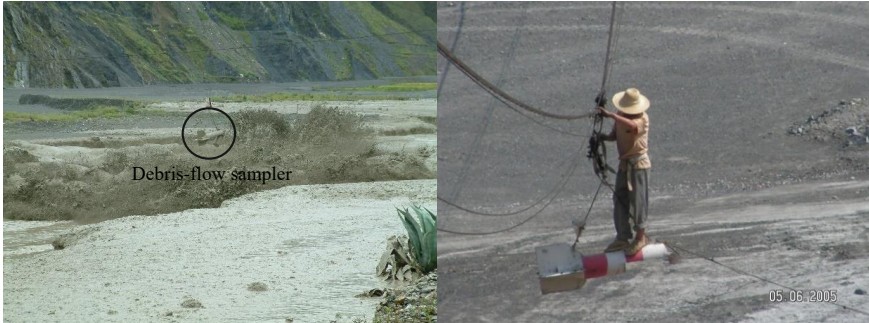

**Figure 9.** Details of debris-flow sampling.

The yield stress and viscosity of the debris-flow slurry (particles smaller than 1.2 mm) were obtained using a Thermo Haake RS600 Rheometer, which employs a concentric cylinder system. This system features an outer cylinder diameter of 43.4 mm and an inner cylinder diameter of 31.44 mm. The debris-flow slurry sample, containing particles smaller than 1.2 mm, had a volume of 50.5 ml. Measurements were conducted using a Z31 rotor, which is 15.72 mm in diameter and 55 mm in height.

**Example of results**

Figure 10a illustrates the grain size distribution of debris-flow deposition samples collected from 1974 to 2024. The bulk densities of the samples range from 2,026 kg/m³ to 2,470 kg/m³, with median grain sizes range from 2 mm and 30 mm. Additionally, Figure 10b depicts the rheological curves of two debris-flow slurries during 2004 to 2004, which have bulk densities range from 1,665 kg/m³ to 2,100 kg/m³. The shear stress increases rapidly at lower shear rates, whereas at higher shear rates, the increase becomes more gradual.

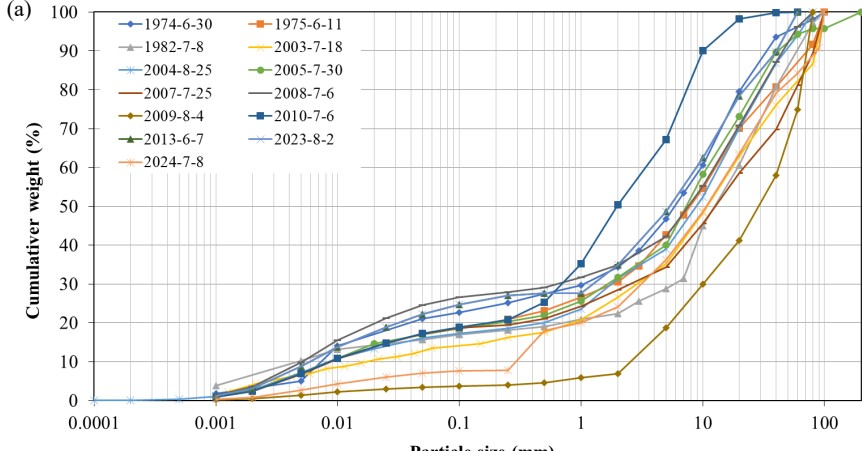


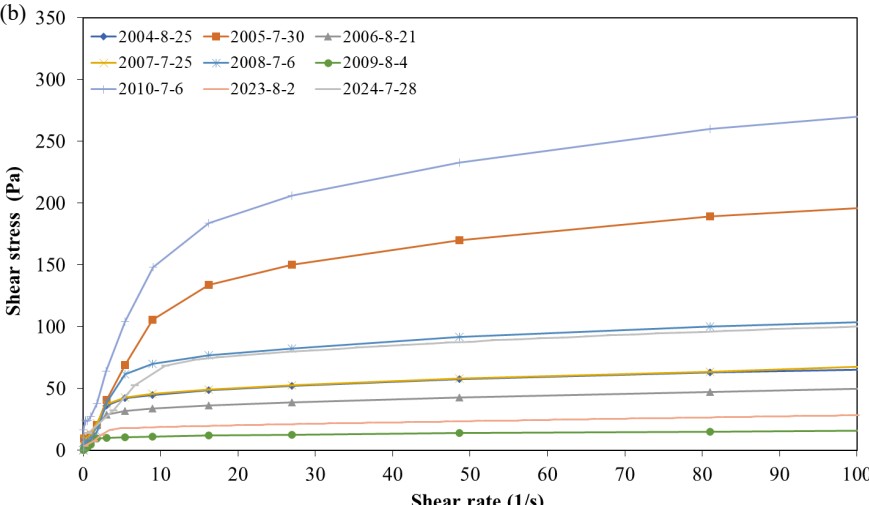


**Figure 10.** (a) Particle size distribution of debris flow surges occurred during 1974 to 2024 and (b)
shear stress of debris-flow slurry occurred during 2004 to 2024.
**3.1.4 Debris-flow video footage**
Since 2020, high-resolution cameras (3840 × 2160 at 25 fps) have been installed on both sides of the
debris-flow channel, including the observation section (Fig.1a). Data is transmitted in real-time through
a wireless relay network and can be viewed online via a mobile client APP. These cameras capture real-
time footage of debris-flow movements, which can be used for research of surge flow formation, flow
state assessment, and others.
**3.1.5 Cross section elevation**
**Measurement**
To investigate the erosion and deposition resulting from the debris flows, the elevation of various
cross-sections was measured after the event. The cross-section elevation was measured using different
methods over several decades. From 1962 to 1966, a level gauge was utilized for measurements. In the



year 1999 and 2010, measurements were conducted using a total station theodolite. Starting in August
2010, a Real-Time Kinematic (RTK) surveying instrument was employed to measure the cross-section
elevation.
**Example of results**

Figure 11 presents the change of elevation of cross section M3 from 1999 to 2024. The M3 section
is located at the debris flow observation section (Fig.1a). Due to the extremely strong siltation of debris
flows, the channel has been raised by 11 m in the past 25 years, with an average annual sedimentation
height of 0.44 m. The main channel gradually shifted to the right bank and has become narrower and
shallower.

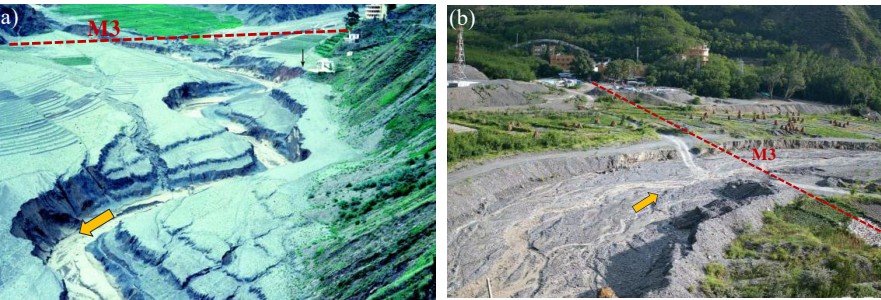


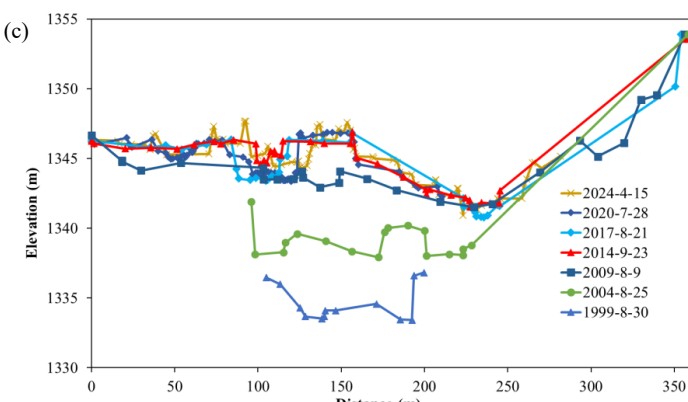


**Figure 11. (a)** Location of cross section M3 in the 1990s (Photo by R.J. Janda), (b) location in 2024
and (c) Change of elevation of cross section M3 from 1999 to 2024 (from right bank to left bank).

**3.2 Atmosphere**
**3.2.1 Rainfall**
**Measurement**

Before 2006, rainfall was measured using a siphon rain gauge with a measurement range of 0.1 mm
to 10 mm and an error margin of ±0.05 mm. This gauge could measure precipitation intensities from 0.1
mm/10 min to 40 mm/10 min. From 2006 to 2023, rainfall measurements were conducted using a tipping-
bucket rain gauge (Fig.12a), which had a range of 0-4 mm/min, a minimum measurement increment of
0.1 mm, and a maximum allowable error of ±4%. In 2024, the rainfall was monitored by piezoelectric
rain gauges (Fig.12b), which had a range of 0-200 mm/h, a minimum measurement increment of 0.1 mm,
and a maximum allowable error of ±4%.





Before 2024, the rainfall data are stored in a logger system that records rainfall time. Every 3 months,
data are retrieved from the logger, the rain gauge is cleaned if needed (leaves, spider web). The 2024
rainfall data measured by piezoelectric rain gauge was transmitted wirelessly via 4G signals. With a
dome-type sensor area, this new type of rain gauge requires no maintenance.
**Example of results**
Detailed records of rainfall allow for deep research into the imitation mechanism of debris flows. An
example of rainfall measured at minute intervals from 01:51 (UTC+8) to 23:00 (UTC+8) on 28th July
2024, at Lijiayakou rain gauge, is shown in Fig. 12c. The debris flow occurred at 11:25, when the
cumulative rainfall reached 41.2 mm and rainfall intensity computed over 1 min reached 31.6 mm/h.
Rainfall records are also aggregated and analyzed on daily scale. Figure 12d illustrates the monthly
distribution of rainfall at the Mayiping rain gauge over the period from 2013 to 2024. With a total annual
rainfall of 827 mm, the heaviest rainfall occurs between June and September.

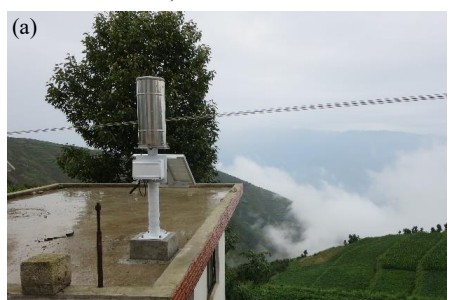
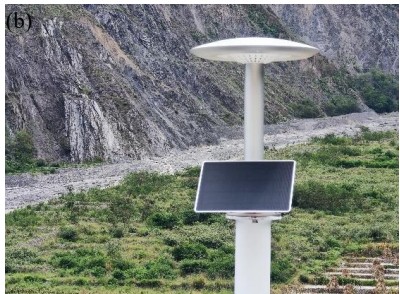


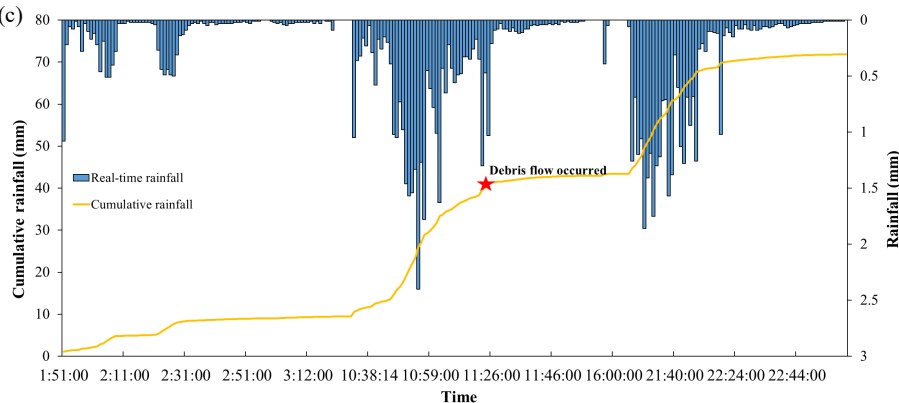


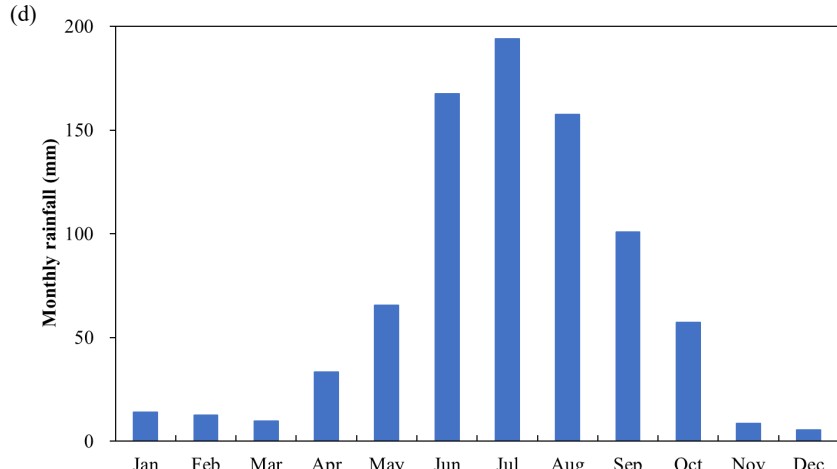

**Figure 12.** (a) Photo of tipping-bucket rain gauge, (b) photo of piezoelectric rain gauge, (c) real-time and cumulative rainfall at Lijiayakou rain gauge on July 28th, 2024 and (d) Monthly rainfall at the Mayiping rain gauge (rainfall data are averaged over period 2013-2023).

### 3.2.2 Meteorology

**Measurement**

Early in the year 1965, meteorological data for Xiadawa included measurements of air pressure, temperature, relative humidity, soil temperature, wind speed, and rainfall. A Cumulus automatic weather station manufactured by ELE International was installed in 2004 and 2005 at the Observation station and Mayiping Station, respectively (See location in Fig. 1a). It includes soil temperature, air temperature, air humidity, wind direction and speed, radiation and pressure. Table 1 summarizes the type of instruments, record interval, accuracy and resolution. Three climatic stations were installed at three locations vary in altitude within the Xiaojiang River Catchment (Nideping station coordinates are 103.11528° and 26.264444°; elevation is 1132 m; Daduo station coordinates are 103.0774° and 26.240339°; elevation is 2030 m; Yinmin station coordinates are 103.01156° and 26.26244°, elevation is 3045 m). The station includes air temperature, air humidity, illuminance, soil moisture and temperature at different depths. Data are stored in a logger system and data are retrieved from the logger every six months.

**Example of results**

The Xiaojiang River Catchment is a typical dry-hot valley region in southwest China, characterized by a prolonged dry season with low precipitation and high evaporation rates (Jiang et al., 2024). Figure 13a illustrates the temporal variation in air temperature at three stations located at different altitudes within the catchment from October 10th to December 13th, 2018. The temperature increases as altitude decreases, with an average temperature difference of 10.4°C between Nideping and Yinmin.

### 3.2.3 Soil moisture

**Measurement**

In 1966, soil moisture at a runoff plot (location see Fig.1a) was manually sampled and analyzed using the oven-drying method. From 2017 to 2024, MS-20 sensors were used to monitor soil moisture and temperature at depths of 10 cm, 20 cm, and 30 cm at three locations with wide-graded gravelly soil at Jiangjia Ravine. These sensors provide precise soil moisture measurements across a 0-100% range and temperatures from -40°C to 80°C. The accuracy of moisture measurements varies: ±2% within the 0-53%

range and ±4% for values exceeding 53%, up to 100%. Additionally, the sensors offer a temperature
measurement accuracy of ±0.4°C.
**Example of results**

Figure 13b illustrates the temporal variation in soil moisture content at different depths at Point 3 on
the right bank of Section D3 (location see Fig.1a), measured on July 28th, 2024. The moisture content in
the near-surface layer (10 cm depth) was lower than in the deeper layers, with an average difference of
11%. Additionally, the soil moisture content at 20 cm depth was higher than at 30 cm depth. This trend
of change may be caused by preferential flow, which were affected by soil porosity characteristics and
soil texture.

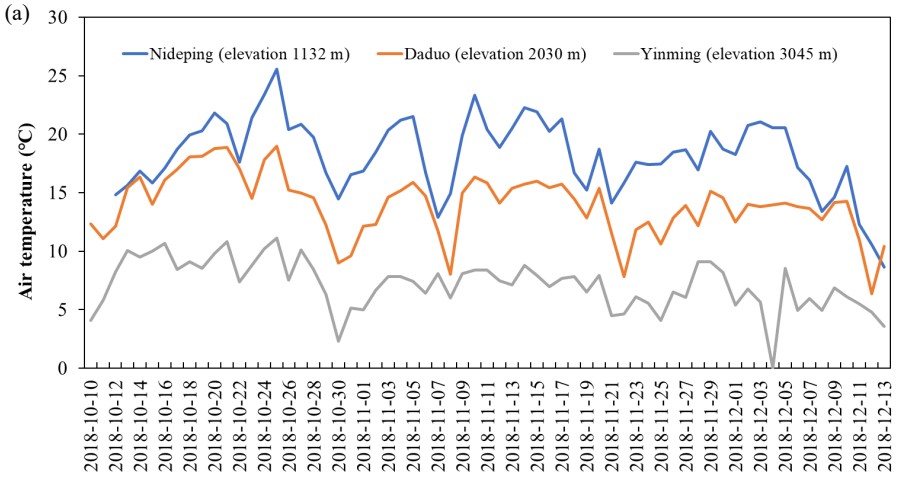


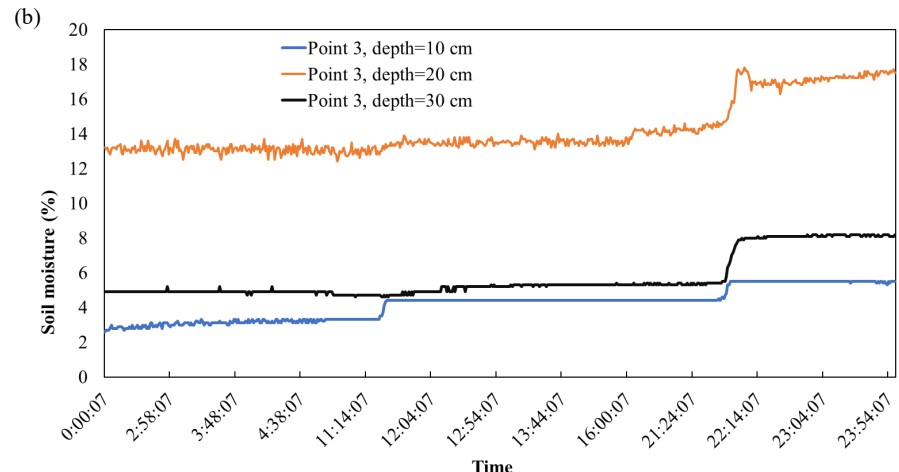


**Figure 13.** (a)Temporal variation of air temperature at Nideping, Daduo, and Yinmin stations from
October 10th to December 13th in 2018 and (b)soil moisture of the right bank of Section D3 at Jiangjia
Ravine on July 28th, 2024.

## 3.3 Runoff and sediment in runoff plot

**Measurement**

Naturally restored runoff plots and Leucaena leucocephala runoff plots were built near DDFORS in 2005 (Fig.14). The runoff plots are 20 m by 5 m in size, and the slope is approximately 22°. In terms of vegetation coverage, the Leucaena leucocephala runoff plots include Leucaena leucocephala (Lam.) de Wit, Leucaena leucocephala (Lam.) de Wit and Agave sisalana Perr. ex Engelm, and Leucaena latisiliqua (L.) Gillis with the crown removed. The naturally restored runoff plots are categorized into forest land dominated by Leucaena leucocephala, agricultural land, and wasteland. These plots have not undergone any artificial interventions, and the data are for reference only. Runoff volume is obtained by measuring the flow volume in the iron bucket. Sediment concentration is obtained by the same method as mentioned above.

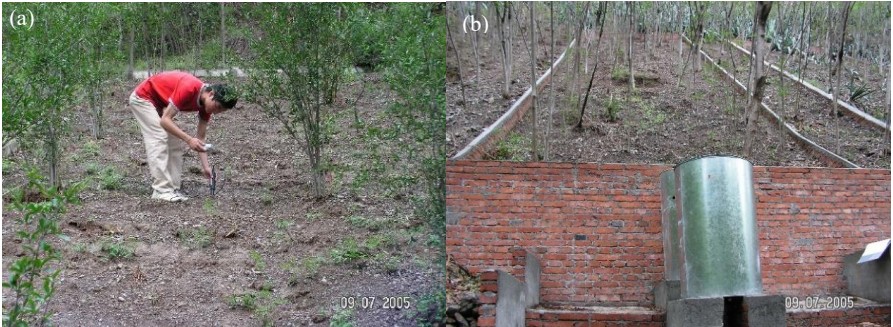

**Figure 14.** Photos of runoff plot: (a) agricultural land runoff plot.(b) forest land runoff plot

## 3.4 Suspended sediment at Jiangjia Ravine and Xiaojiang River

**Measurement**

In 1966, the runoff discharge at Jiangjia Ravine in Baishapo and Xiadawa was estimated based on manually measured flow area and velocity. From 2004, the suspended-sediment concentration and grain size distribution were measured in Xiaojiang River at different locations. From 2012, the sediment concentration and sediment grain size distribution of runoff were measured at Jiangjia Ravine. A water sample was manually collected with a volume of 500 ml, and then sediment contained within the sample was filtered, dried, and weighed. Sediment concentration was determined by comparing the weight of the sediment to the total sample weight. Particle size analysis of the sediment was conducted using a Malvern laser particle size analyzer (Model MS2000).

**Example of results**

An example of the suspended sediment concentration time series for the Xiaojiang River, obtained from May 10th to December 3rd 2007 at the Gele station (103.0611438°, 26.53562376°), is presented in Fig.15a. The highest concentration, 43 kg/m³, was observed on 13th July. The concentration varied significantly between July 11th and September 23rd, a period characterized by frequent debris flows in the Xiaojiang River catchment, which transported large amounts of sediment to the main river. Figure 15b also illustrates the variation in suspended sediment concentration at Jiangjia Ravine between June 9th and August 20th, with the highest concentration, 628 kg/m³, observed on June 30th. Concentrations exceeding 628 kg/m³ were consistently recorded following debris-flow events when floods carried additional sediment and caused a significant increase in sediment concentration.

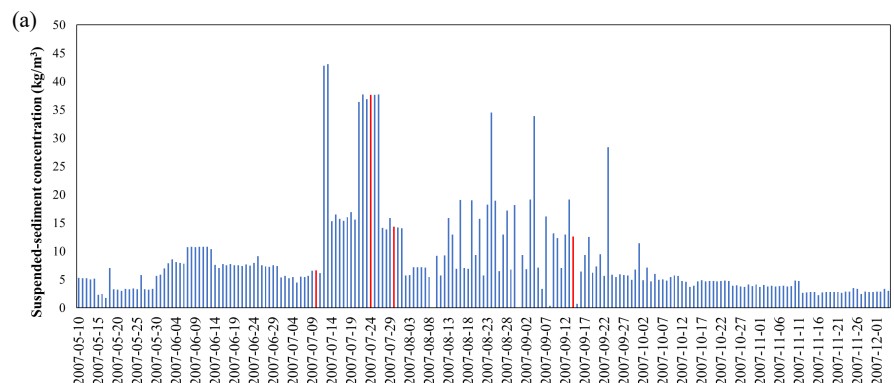

**Figure 15.** Suspended sediment concentration variation: (a) Xiaojiang River at Gele station in 2007.
(b) Jiangjia Ravine in 2015. The red bars represent water samples collected after the debris-flow event
at Jiangjia Ravine.

## 4. Examples of studies and open questions

This debris-flow dataset is valuable for analyzing the dynamic behavior, deposition, and erosion
characteristics of debris flows. Several studies are presented to demonstrate the dataset's potential and to
elicit the open questions.

### 4.1 Debris-flow dynamic characteristics

The comprehensive records of debris-flow depth, velocity, density, and discharge enable the analysis
of dynamic characteristics of debris flows and the comparison of various computational models (Wu,
1987; Hu et al., 2011; Hu et al., 2013; Tian et al., 2014; Zhu et al., 2020; Chen et al., 2023; Chen et al.,
2024). In the early 1980s, based on observational data at Jiangjia Ravine, the Chinese researchers
modified the Manning formula for velocity and resistance calculation of viscous (high solid-
concentration) debris flows. These formulas have been widely applied in the design of debris-flow
mitigation structures across China (Chen and Wang, 1983; Kang, 1984, 1987, 1990,2004; China
Geological Disaster Prevention Engineering Association, 2018a, 2018b; Kwan, 2012). Additionally,
Zhang (1993) reported that the impact pressure of debris flows obeys the conservation of momentum
(proportional to the unit weight and the square of the flow velocity), while the impulse load of boulders
within debris flows varies with the stiffness of the exposed elements.



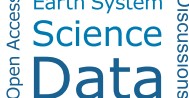

The flow-depth time history (e.g., Fig. 6a) provides solid evidence for verification of dynamic models.
Chen et al. (2024) adopted the surge-depth hydrographs measured by ultrasonic sensors (1999-2001) to
quantify the eroded deposition depth of surge flows (see Fig. 4c). For surge flows with erosion-deposition
propagation, significant downward erosion potential is confirmed. Therefore, the total momentum of
surge flow not only originates from the apparent surge front, but also includes the momentum within the
eroded deposition layer. The revealed erosion pattern and hidden momentum in debris-flow surges may
improve the reliability of debris-flow risk assessment.
Based on 5,085 debris-flow measurements collected from Jiangjia Ravine between 1990 and 2001,
along with 1,035 measurements from catchments across Asia, America, Europe, and Oceania, Du et al.
(2023) established a criterion for distinguishing debris flows, hyperconcentrated flows, and stream flows.
Their findings indicate that sediments in hyperconcentrated flows and stream flows are primarily
supported by viscous shear and turbulent stresses, whereas grain collisional stresses play a dominant role
in debris-flow dynamics. The study identifies flow discharge and sediment flux as key factors
differentiating debris flows, hyperconcentrated flows, and stream flows.

### 4.2 The non-hydrostatic pore fluid pressure

Current research, integrating field observations (McArdell et al., 2007; Nagl et al., 2020) and flume
experiments (Cassar et al., 2005; Iverson et al., 2010; Kaitna et al., 2014; Song et al., 2021, 2023), has
recognized the non-hydrostatic pore fluid pressure, notably excess pore fluid pressure, in debris flows.
However, direct measurement of basal normal stress, shear stress, and pore fluid pressure at Jiangjia
Ravine remains impractical, due to challenges posed by erosion, deposition, and channel migration. To
circumvent these limitations, Chen et al. (2023) proposed a simplified analytical method to estimate
debris-flow liquefaction. By assuming steady-state flow on a 3.7° slope and neglecting particle collision-
induced resistance, they attributed flow resistance primarily to Coulomb friction regulated by pore fluid
pressure and liquid-phase viscosity. Analysis of 93 debris-flow events (1999-2017) revealed near-
liquefaction conditions, with degree of liquefaction ranging from 0.89 to 0.95. Therefore, with the
consideration of particle collision-induced resistance, the degree of liquefaction would be even higher
(close to unity). This finding demonstrates that analysis relying on hydrostatic pore fluid pressure
assumptions to infer grain-contact stresses (i.e., effective stresses) are invalid. Such approach, initially
introduced by Iverson (1997) to calculate dimensionless parameters (e.g., Savage number, friction
number) for debris flows, have been largely abandoned in light of direct pore fluid pressure and effective
stress measurements (Iverson et al., 2010). As a result, we caution against the derivation of flow regime
characteristics (by dimensionless parameters) from observational datasets lacking direct basal stress
measurements.

### 4.3 Surge flow characteristics and surge flow formation

Debris flows at Jiangjia Ravine propagate as surge waves (roll waves), comprising dozens to
hundreds of intermittent surges. Discharge values within a surge series (1966-2004) follow an
exponential cumulative distribution, with exponents scaling as a power-law function of peak discharge
(Liu et al., 2009). This implies that the mean discharge evolves dynamically during the surge sequence,
ultimately decaying via a power-law relationship. Surge velocities in 1994 conform to a Weibull
distribution, characterized by stable parameters across events, where the shape parameter correlates with
mean velocity (Li et al., 2012). Similarly, the peak discharge-frequency relationship for surges (1987-
2004) adheres to a Weibull distribution (Chen et al., 2011), while the magnitude-cumulative frequency
relationship aligns with either an exponential function or a linear logarithmic transformation (Liu et al.,



2008; Gao et al., 2019). Notably, discharge fluctuations within individual events span four orders of
magnitude, with sediment transport variability diminishing in a power-law manner as surges progress
(Liu et al., 2023).
The debris-flow surge waves carry much higher momentum flux than the continuative flows in the
same debris-flow event, posing higher destructive potential to the infrastructures. The formation of surge
flows is currently one of the hot research topics of debris-flow dynamics. The discontinuous sediment
supply is regarded as a potential explanation of surge flows. The formation of intermittent surges arises
from shallow slope failures distributed across broad source areas rather than discrete large landslides.
These failures can be modeled using a Pareto-Poisson process (PPP model), where the intensity
parameter is governed by rainfall dynamics in source regions (Guo et al., 2023). Through captured
processes by video camera at Chalk Cliffs, Kean et al. (2013) attributed the surge flow formation by
upstream variations in channel slope. The low-gradient sections act as "sediment capacitors," temporarily
storing incoming bed load transported by water flow and periodically releasing the accumulated sediment
as debris surges. This "store-release" pattern is also observed in the small branch gullies of Jiangjia
Ravine. However, this pattern may not explain the surge behavior in the main channel, which is
characterized by high degree of liquefaction. Further understanding of the surge flow formation, on one
hand, relies on the intense and precise instrumentation in the debris-flow source area (e.g., processes
captured by video cameras), on the other hand, benefits from the development of granular-fluid flow
models (Meng et al., 2022).

**4.4 Debris-flow grain composition**

The effect of particle size on debris-flow behavior is a fundamental question. Previous research
framework adopted specific particle parameters to distinguish clay, silt, sand, and gravel. Based on
particle size distribution data from debris flows in Jiangjia Ravine, Li et al. (2013) developed a
generalized grain size distribution model defined by parameters closely linked to key dynamical
properties, including flow density, velocity, and discharge. Subsequent studies demonstrate that grain
composition imposes power-law constraints on debris-flow surge fluctuations (Li et al., 2014), while
debris-flow density and cumulative sediment yield adhere to a unified sediment size distribution
framework (Wang et al., 2018; Liu et al., 2023;Zhang et al.,2025). Observations reveal a progressive
increase in fine particles and a corresponding decline in coarse particles during intermittent flow
sequences, alongside reductions in median grain diameter, bimodality parameters, and mass density.
viscous particle content in continuous flows increases incrementally, whereas mass density exhibits a
pronounced decrease (Wei and Hu, 2014).

**4.5 Debris flow erosion and deposition characteristics**

Between 1966 and 1973, the debris-flow channel at Jiangjia Ravine experienced substantial erosion
and deposition due to viscous (high solid fraction) debris flows. The average annual scouring depth in
the upper reaches was 2-3 m, while the average annual deposition depth in the lower reaches was 2-2.5
m (Kang, 1997). During this period, a single debris-flow event could erode the channel bed to a depth of
2-3 m, and sometimes even 8-10 m (Li et al., 1983). From 1999 to 2003, the relatively lower magnitudes
of debris flows led to sediment deposition in the majority of the river sections. However, the increasing
annual sediment discharge resulted in a reduction of deposition in the upper reaches and an enhancement
of deposition in the lower reaches (Chen et al., 2005). Wei et al. (2017) reported that, between 1999 and
2009, channel width increased significantly, stabilizing after 2009 when erosion and deposition intensity
decreased. From 2003 to 2014, the gradient of the channel profile experienced a gradual increase,



eventually reaching 0.07, although the rate of increase slowed after 2009. Debris flows were primarily
characterized by siltation in the transportation zone, with variations across different sections. The middle
and upper reaches experienced greater siltation, while the middle and lower reaches underwent more
substantial scouring (Fang et al., 2018). Overall, from 1962 to 2024, the channel has undergone
significant siltation. The underlying causes of this channel aggradation may be related to climate change
induced reduction in debris-flow frequency or long-term geomorphic evolution controlled by the tectonic
activities.

**4.6 Rainfall threshold triggering debris flows**

Detailed rainfall records have enabled the investigation of rainfall threshold in debris-flow
forecasting. Debris flows are primarily triggered by intraday precipitation lasting 6 hours or less, while
antecedent precipitation is not a significant factor,the Intensity-Duration (I-D) thresholds for debris flows
at 50%, 70%, and 90% probabilities have been determined (Zhuang et al., 2015). In contrast, Rainfall
threshold for triggering debris flows do not consistently decrease with increasing antecedent effective
precipitation (AEP). In fact, higher AEP may require more stringent triggering conditions (Zhang et al.,
2023). The relationship between the probability of a debris flow and AEP can be quantified using a
piecewise function. When 10 mm ≤ AEP ≤ 85 mm, the rainfall I-D threshold curves can be described by
an exponential function. This indicates that higher soil water content allows solid material from shallow
landslides to become readily available without prolonged rainfall infiltration, and a high soil water
content in the topsoil facilitates rapid runoff generation (Zhang et al., 2024). Yang et al. (2024)
demonstrated that models using maximum rainfall intensity over short durations and absolute energy are
the most effective predictors for debris-flow occurrence. Adding rainfall duration or antecedent rainfall
to these models further improves their performance.
The intermittent failure of shallow landslides makes the rainfall-induced debris flows at Jiangjia
Ravine even complicated. Guo et al. (2020) revealed that a debris flow is the product of a rainfall-induced
"normal" or "abnormal" hydrological process involving the supply of soil materials. They argued that
although various types of debris flows may require similar rainfall conditions, the threshold for debris-
flow initiation can be expressed as a power function, and that the threshold could be expressed as power
function. Debris flows are more closely related to rainfall patterns than to rainfall amounts, typically
occurring within 6 hours of a rainfall event characterized by high mean intensity and short duration.
However, the formation type, process, and discharge of debris flows show no direct relationship with
rainfall amount. These findings highlight the significance of soil supply in debris flow formation through
random disturbances to the normal hydrological process. Additionally, Guo et al. (2021) demonstrated
that interpolation errors decrease with higher rainfall amounts, while the representativeness of rain
gauges diminishes when the distance between gauges exceeds 3 km. The influence of climate change
such as changes in precipitation intensity and frequency, and changes in temperature on the triggering
mechanisms of debris flows at Jiangjia Ravine requires further analysis.
For a more comprehensive list of data-based research papers, please refer to
https://nsl.imde.ac.cn/en/publ/article_2020/.

## 5. Data availability

The debris flow, atmospheric, runoff, and sediment data, including the suspended sediment
concentration and particle size in the Xiaojiang River, are publicly accessible through the National
Cryosphere Desert Data Center (NCDC) (https://www.ncdc.ac.cn/). The dataset is organized into 11
categories: dynamic parameters of debris flow, seismic data, particle size distribution of debris flow



sediment, yield stress and viscosity of debris-flow slurry, debris-flow video footage, cross-sectional
elevation of the debris flow channel, rainfall, meteorological data, soil moisture, runoff and sediment
data in runoff plots, and suspended sediment concentration (Table 3). Detailed descriptions of the
measurement methods, data collection locations, and data processing techniques are included for each
category. All data files are provided in China Standard Time (CST, UTC+8).
This debris-flow dataset will be updated using newly collected measure data at a one-year interval.
The updated dataset will continue to be released freely and publicly on the National Cryosphere Desert
Data Center (NCDC).
**Table 3** Dataset categories and open repository.

| Dataset | Data DOI | Reference |
|---|---|---|
| Debris-flow kinematic data | http://dx.doi.org/10.12072/ncdc.ddfors.db6803.2025 | Song et al., 2025a |
| Seismic data of debris flow | http://dx.doi.org/10.12072/ncdc.ddfors.db6804.2025 | Song et al., 2025b |
| Particle size distribution of debris flows | http://dx.doi.org/10.12072/ncdc.ddfors.db6721.2025 | Song et al., 2025c |
| Rheological data of debris-flow slurry | http://dx.doi.org/10.12072/ncdc.ddfors.db6720.2025 | Song et al., 2025d |
| Debris-flow video | http://dx.doi.org/10.12072/ncdc.ddfors.db6807.2025 | Song et al., 2025e |
| Cross-sectional measurement data | http://dx.doi.org/10.12072/ncdc.ddfors.db6719.2025 | Song et al., 2025f |
| Meteorological data | http://dx.doi.org/10.12072/ncdc.ddfors.db6805.2025 | Song et al., 2025g |
| Rainfall data | http://dx.doi.org/10.12072/ncdc.ddfors.db6716.2025 | Song et al., 2025h |
| Soil moisture and temperature data | http://dx.doi.org/10.12072/ncdc.ddfors.db6718.2025 | Song et al., 2025i |
| Sediment concentration and grain size distribution data | http://dx.doi.org/10.12072/ncdc.ddfors.db6802.2025 | Song et al., 2025j |
| Observation data at runoff plots | http://dx.doi.org/10.12072/ncdc.ddfors.db6806.2025 | Song et al., 2025k |

## 6. Conclusion
We present a comprehensive dataset of debris-flow and hydrometeorological observations collected
from 1961 to 2024 at Jiangjia Ravine. This collection encompasses detailed measurements of kinematic
parameters (such as debris-flow velocity, depth, and surge discharge) and physical-mechanical
parameters (including particle size, yield stress, and viscosity), along with seismic data. In addition, it
features extensive records of rainfall, soil moisture, and suspended sediment concentrations at the
catchment scale. The breadth of this documentation supports rigorous analyses of debris-flow initiation,
transport, and deposition, while highlighting the crucial role of hydrometeorological conditions
particularly rainfall in triggering debris flows. Furthermore, the suspended sediment concentration data
provides valuable insights into sediment transport within mountainous watersheds prone to frequent
debris-flow events. The dataset is also useful for the research of influence of climate change on debris-
flow occurrence, tectonic-disaster-geomorphic evolution. Overall, this dataset constitutes a vital resource
with significant potential to advance both theoretical and practical research on debris-flow processes.
## Author contribution
DS designed the framework of this research; LW, XG and HT performed the data analyses and wrote



the manuscript; ZK, JZ, GO, PC, FW, KH, LS, GZ, DS, WZ, YZ contributed to data collection; YH and
XL installed and maintained the instruments and infrastructures.

**Competing interests**
The corresponding author declares that none of the authors has any competing interests.
**Acknowledgements**
This work has been financially supported by the Science and Technology Research Program of
Institute of Mountain Hazards and Environment, Chinese Academy of Sciences (Grant No. IMHE-
JCCX-02), the National Natural Science Foundation of China (Grant No. 42477193, 52409109), National
Cryosphere Desert Data Center (Grant No. E01Z790201). We would like to thank Jishan Wu, Pinghua
Hu, Jingwu Chen, Renwen Yang, Yuyi Wang, Jingri Chen, Huifang Diao, Guisheng Luo, Mingfu Ye,
Xingwen Wu, Gang Xiong, Yuzhang Wang, Kai Wang, Ningsheng Chen, Xuelan Liu, Shucheng Zhang,
Shunli Chen, Guoqiang Tu, Youfu Zhang, Shufen Hu, Zunlan Cheng, Yong You,Jinshan Zhangand all
others colleagues of Dongchuan Debris Flow Observation and Research Station (DDFORS), Chinese
Academy of Sciences, who contribute to the field observation of Jiangjia Ravine debris flows since the
year 1961.



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
