# Peer review of "A long-term dataset of debris-flow and hydrometeorological"

_Earth System Science Data, 2025_

## Author Comment (AC1)

A long-term dataset of debris-flow and hydrometeorological observations from 1961 to 2024 at Jiangjia Ravine, China, [Paper # essd-2025-190]

**Reply to reviewers' comments**

(*C and R denote comment and reply, respectively*)

**Reviewer 1**

**C0:** General comments:

This is a rich and unique dataset, which can potentially very useful for future research. I am not aware of anything similar in landslide/debris-flow research. Therefore, this could be a seminal contribution in terms of open data research. The authors did a great job in collecting and synthesizing the different datasets and translating descriptions and headers to English. All links to the dataset work and the data can be downloaded. Before publication, I think some figures and example data could be presented differently. Furthermore, some more details on assumptions and calculations could be added (see comments below). In my opinion, it can be published in ESSD if these points are addressed.

**R0:** Thank you for the constructive comments. The dataset has garnered a total of 18,186 views and 3,705 downloads since its release on April 2025. We are delighted to contribute to the fundamental research on debris flows. We have made revisions on a point-by-point basis. Please see our detailed reply to comments below.

**C1:** L42-L57: this is a nice list of monitoring installations, but would be much more accessible if you could put them in a map or table.

**R1:** Thank you for the suggestion. We have added a table that includes the installation year of each monitoring station, maximum altitude, drainage area, and main channel length of each catchment.

*Table 1 Characteristics of typical monitoring sites*

| *Catchment* | *Country-Region* | *Year of installation* | *Maximum altitude (m asl)* | *Drainage area (km²)* | *Main channel length (km)* | *Reference* |
|---|---|---|---|---|---|---|
| *Lattenbach* | *Austria* | *2002* | *2930* | *5.3* | *5.2* | *Hübl and Moser, 2006;Hübl and Kaitna, 2010; Hübl et al., 2017* |
| *Wartschenbach* | *Austria* | *1995* | *2113* | *2.7* | *3.6* | *Fuchs et al., 2012* |
| *Illgraben* | *Switzerland* | *2000* | *2716* | *11.7* | *5.5* | *McArdell et al., 2007; Berger et al., 2011; Hirschberg et al., 2021;Aaron et al.,2023; Raffaele and Jordan, 2024* |
| *Dorfbach* | *Switzerland* | *1993* | *4545* | *5.6* | *3.2* | *Willi et al.,2015* |
| *Spreitgraben* | *Switzerland* | *2009* | *3263* | *4.7* | | *Tobler et al., 2014* |
| *Moscardo* | *Italy* | *1989* | *2043* | *4.7* | *2.76* | *Marchi et al., 2002; Blasone et al., 2015* |
| *Acquabona* | *Italy* | *1997* | *2667* | *0.3* | *1.6* | *Berti et al., 1999; Genevois et al., 2000;* |
| *Gadria* | *Italy* | *2011* | *2945* | *6.3* | *3* | *Comiti et al., 2014; Theule et al., 2018* |

| | | | | | | |
|---|---|---|---|---|---|---|
| *Manival* | *France* | *2010* | *1738* | *3.6* | *1.8* | *Theule et al., 2015; Bel et al., 2015* |
| *Réal* | *France* | *2010* | *2090* | *2.3* | *2.6* | *Navratil et al., 2013* |
| *Rebaixader* | *Spain* | *2009* | *2475* | *0.53* | *1.4* | *Abancó et al., 2014* |
| *Portainé* | *Spain* | *2015* | *2439* | *5.72* | *5.7* | *Hürlimann et al., 2013* |
| *Kamikamihorizawa* | *Japan* | *1970* | *2455* | *0.8* | *2.5* | *Suwa et al., 1993; Okano et al., 2012; Ikeda et al., 2023* |
| *Chalk Cliffs* | *USA* | *2004* | *3140* | *0.3* | *1* | *Pierson ,1986; Coe et al., 2010; McCoy et al., 2011, 2012* |
| *Shenmu* | *Taiwan, China* | *2002* | *2850* | *72.2* | *17.7* | *Yin et al., 2011* |
| *Yushui Stream* | *Taiwan, China* | *2018* | *2756* | *12.3* | *7.31* | *Liu and Wei, 2024* |
| *Jiangjia Ravine* | *China* | *1961* | *3269* | *48.6* | *13.9* | |

**C1:**L44: I don't think Erlenbach produces debris flows, but bed load transport. Please double check

**R1:** Thank you for the kind reminder. We have checked the characteristics of the Erlenbach Torrent and confirmed that bedload, not debris flow, is transported in this catchment. Therefore, we have removed this catchment from the manuscript.

**C2:** Figure 3: this is a bit small. Maybe you can flip it by 90° and fill a page?

**R2:** We have flipped it by 90° and fill a full page. Please see the revised manuscript.

**C3:** Figure 4: I would add the channel bed to panel c. Is flow depth *H+h* or only *H*?

**R3:** The flow depth is *H*. According to your suggestion, we added the channel bed to the figure as follows

[Figure]

**Figure 4.** (c)illustration of flow depth

**C4:**L220: please add that you assume a rectangular channel cross section.

**R4:** We added this assumption in the revised manuscript (Page 9, Line 221-223):

*"The discharge, volume, sediment concentration, sediment volume, and sediment transport rate of debris-flow surge were determined based on the observational parameters. A rectangular channel cross section was assumed."*

**C5:**L224: why *T/2*?

**R5***:* The following figure shows the debris depth hydrograph, measured by an ultrasonic level meter, as a debris flow surge passes through the cross-section. The debris depth clearly exhibits a triangular shape, with *H* representing the maximum depth at the front (the apex of the triangle) in Equation (1). Given the peak depth *H* and surge duration *T* are known, the total discharge can be simplified as:

$$Q = \frac{1}{2} \times V \times B \times H$$

Therefore, when calculating the discharge for surge-type debris flows, the factor $T/2$ is applied.

[Figure]

Flow depth hydrograph measured by ultrasonic level meter as a debris flow surge passes through the cross-section on August 8, 2000 (http://dx.doi.org/10.12072/ncdc.ddfors.db6803.2025).

We have added this information in the revised manuscript (Page 10, Line 227-232):

"*Surge flow volume* $W_c$ *(m³) was calculated as following:*

$$W_c = Q \times \frac{T}{2} \quad \text{for surge flow} \tag{2}$$

$$W_c = Q \times T \quad \text{for continuative flow} \tag{3}$$

*where T is the record time (duration) of debris-flow surge (s), which is the time for surge front minus the time for surge rear. Flow depth hydrograph of surge flow exhibits a triangular shape, so the factor 1/2 is applied in the calculation.*"

**C6**:L225: please define how you differentiate surge and continuative flow

**R6**: We have added the definition in the revised manuscript (Page 4, Line 158-161):

"*These surges, known as surge flows, are characterized by a distinct head and body. Surge flows are typically separated by periods of flow interruption or quiescence. In contrast, when a debris flow persists for an extended period without noticeable surge features(such as a pronounced head), it is classified as a continuative flow.*"

**C7**:L230: what is $r_s$?

**R7**: We apologized for this typo. We have corrected it to $\gamma_s$ (density of sediment), no new parameter is introduced.

**C8**:L234: I don't get the logic with the subscripts c and s. Why is sediment volume $W_s$, but sediment transport rate $Q_c$ and not $Q_s$?

**R8**: In the history, the sediment transport rate $Q_c$ was used in some published literature of our observation. To maintain consistency with this previously published literature, this symbol has been retained in this dataset.

**C9**: Figure 6: consider using other colors because in the previous figures you use these to differentiate surge types

**R9**: Thank you for your suggestion. We have redrawn the two figures as follows:

[Figure]

[Figure]

**Figure 6.** (a) Variation of flow depth, velocity and (b) discharge, volume of debris-flow surges occurred on July 16th, 1999.

**C10**: Figure 7: these are huge inter-annual variabilities and the catchment seems inactive now. Is this correct? Can this variability be explained with interruptions of systematic monitoring?

**R10**: As shown in Figure 3, debris flows indeed occurred in 1968, 1969, 1970, 1971, 1972, 1973, 1976, 1977, 1978, 1979, 1980, 1985, 1986, 2015, 2019, and 2020. However, in these years, the debris flows did not reach the monitoring section, resulting in no observational data (these years are also annotated in Figure 7). Partial surge data are missing for the events in 1974 and 1975, and no total sediment transport data is available for these years.

Overall, the frequency and magnitude of debris flows have significantly declined in recent decades. However, the events in 2023 and 2024 still recorded significant sediment transport volumes of 52,682 m³ and 31,450 m³, respectively, which remain substantial for mountainous watersheds. By the way, we are currently analyzing the relationship between climate change and sediment transport, to reveal the key factors that control the declined rate of sediment.

We have added the description in the revised manuscript (Page 12, Line 273-276):

"*Overall, the frequency and magnitude of debris flows have significantly declined in recent decades. However, the events in 2023 and 2024 still recorded significant sediment transport volumes of 52,682 m³ and 31,450 m³, respectively, which remain substantial for mountainous watersheds. The decreasing trend may be a result of climate change.*"

[Figure]

**Figure 7.** Variation of sediment transported by debris flow recorded from 1966 to 2024. * denotes debris flows occurred without observational data.

**C11**:L384: why is rainfall and meteorological data separated but still rainfall is mentioned here again?

**R11**: The meteorological data table for 1965 is sourced from a published dataset and primarily includes meteorological observation data, with only six days of rainfall data. To maintain consistency with the published data, the rainfall data was not deleted.

**C12**: Figure 12d/13/15: I understand that you want to show what your data looks like and this makes sense for data of debris flow events, but I don't think this is very informative for long-term meteo data. I would consider bar plots (like 12a) showing monthly mean and error bars with e.g. min/max values from your observation period.

**R12:** Long-term time series data would be more informative, but the figures presented in the manuscript only show short-term data. In fact, except rainfall, the meteorological and sediment concentration data in the dataset are relatively limited, making it difficult to construct statistical charts of long time series. Figure 12 has been revised according to your suggestions, and Figure 13b has been removed.

We also added the following description in the manuscript (Page 20, Line 388-391):

*"The boxplot Fig. 13d illustrates the distribution of monthly rainfall throughout the year. Rainfall shows obvious seasonality, peaking during the rainy season (June to September) and reaching its lowest levels during the dry season (November to March)."*

[Figure]

**Figure 13**. (d) statistical box plot of monthly rainfall at the Mayiping rain gauge (rainfall data are averaged over period 2013-2023). The points above the box plot denote extreme monthly rainfall.

**C13**: Data repositories: on the webpage in the box to the right « how to get the data » it says « download via FTP » instead of « http »

**R13:** All the data can be downloaded directly or via FTP. For large datasets, it is recommended to use FTP (as shown in the image below). Both download methods are available.

[Figure]

**C14**: Soil moisture of runoff plot at Jiangjia Ravine in 1966.xlsx: should relative water content be in unit g?

**R14:** The relative water content should be expressed in percentage (%). Our apology. We have corrected this in the table.

**C15**: Rainfall data: this seems to be in several data sets (meteorological, rainfall, kinematic). Maybe you could explain the differences in the text where you mention Table 3?

**R15**:As explained in our reply **R11**, the rainfall data (rainfall and debris flow data for 1961 and 1965) in the kinematic dataset were sourced from published paper-based (hard copy) datasets. To maintain consistency, these rainfall data were not removed.

In the meteorological dataset, rainfall was obtained through automated observations at weather stations. To preserve data integrity, the rainfall records remain included in the meteorological tables. The rainfall dataset primarily derives from rain gauges with minute-level resolution.

---

## Author Comment (AC2)

A long-term dataset of debris-flow and hydrometeorological observations from 1961 to 2024 at Jiangjia Ravine, China, [Paper # essd-2025-190]

**Reply to reviewers' comments**

(*C and R denote comment and reply, respectively*)

**C0**: The manuscript presents a long-term dataset of debris-flow and hydrometeorological observations from 1961 to 2024 at Jiangjia Ravine, China, compiled from the Dongchuan Debris Flow Observation and Research Station (DDFORS). This is a valuable and significant contribution to the field of debris-flow research. Long-term, continuative, and event-specific data of this kind are rare and provide essential support for understanding debris-flow dynamics, temporal trends, and the influence of changing climatic and environmental conditions on debris-flow occurrence and behavior.

However, before the manuscript can be considered for publication, several important points need to be addressed to further improve the clarity, completeness, and usability of the dataset.

**R0**: Thank you for helping us to improve our manuscript with constructive comments. The dataset has garnered a total of 18,186 views and 3,705 downloads since its release on April 2025. We are delighted to contribute to the fundamental research on debris flows. We have made revisions on a point-by-point basis. Please see our detailed reply to comments below.

**C1**: In Figure 7, the sediment transported by debris flow shows considerable inter-annual variability. Could the authors clarify whether this level of variation is consistent with the actual field conditions, and if so, provide a reasonable explanation for these fluctuations?

**R1:** As shown in Figure 3, debris flows indeed occurred in 1968, 1969, 1970, 1971, 1972, 1973, 1976, 1977, 1978, 1979, 1980, 1985, 1986, 2015, 2019, and 2020. However, in these years, the debris flows did not reach the monitoring section, resulting in no observational data (these years are also annotated in Figure 7). Partial surge data are missing for the events in 1974 and 1975, and no total sediment transport data is available for these years.

Overall, the frequency and magnitude of debris flows have significantly declined in recent decades. However, the events in 2023 and 2024 still recorded significant sediment transport volumes of 52,682 m$^3$ and 31,450 m$^3$, respectively, which remain substantial for mountainous watersheds. By the way, we are currently analyzing the relationship between climate change and sediment transport, to reveal the key factors that control the declined rate of sediment.

We have added the description in the revised manuscript (Page 12, Line 273-276):

 "Overall, the frequency and magnitude of debris flows have significantly declined in recent decades. However, the events in 2023 and 2024 still recorded significant sediment transport volumes of 52,682 m$^3$ and 31,450 m$^3$, respectively, which remain substantial for mountainous watersheds. The decreasing trend may be a result of climate change."

[Figure]

**Figure 7.** Variation of sediment transported by debris flow recorded from 1966 to 2024. * denotes debris flows occurred without observational data.

**C2**: The text in Figures 3 and 8b is difficult to read due to low resolution. It is recommended to improve the image quality and enhance the readability of the labels and annotations in these figures.

**R2**: Thank you for your suggestion. We have redrawn the figure as follows:

[Figure]

**Figure 8**. (b) the time-domain seismic signal and time-frequency characteristic curves of the debris-flow event on July 28th, 2024.

**C3**: In Section 3.1.4, it would be helpful if the authors could provide a few examples of debris-flow monitoring images or video frames captured by the high-resolution cameras. This would better illustrate the application of these instruments in actual debris-flow observation.

**R3**: Thank you for your suggestion. We have added images of the first debris-flow surge on coarse bed, multiple debris-flow surges, continuative flow, and hyperconcentrated flow, all captured on July 28th, 2024.

[Figure]

[Figure]

**Figure 8.** Debris-flow video frames captured on July 28th, 2024.: (a) first debris flow surge on coarse bed, (b) surge flow, (c) continuative flow, and (d) hyperconcentrated flow.

**C4**: In Figure 13b, the soil moisture distribution at different depths indicates that the middle layer (20 cm) has the highest moisture content, while both the surface layer (10 cm) and lower layer (30 cm) show lower moisture values during the observation period. The authors are encouraged to verify whether this distribution pattern is reasonable and discuss under what conditions the soil moisture at the middle layer would exceed that of both the upper and lower layers.

**R4**: This trend may be attributed to preferential flow, which is influenced by soil porosity characteristics and texture (cracks). However, we were unable to determine a plausible explanation for this phenomenon. Consequently, we have removed this figure and its accompanying description. We have also documented this modification in the data explanation document.

**C5**: Regarding data reliability, it would strengthen the manuscript if the authors could clarify whether any cross-validation or data consistency checks were performed to verify the validity of the long-term observations.

**R5**: Most datasets, including debris-flow kinematic data, particle size distribution of debris flows, rheological data of debris-flow slurry, cross-sectional measurement data, sediment concentration, grain size distribution data, and observation data from runoff plots, were obtained through manual observation. These data are considered to have relatively small errors. Additionally, we performed calibration and verification of rain gauges and weather stations in 2021. Some of the results are shown in the following figures, indicating high reliability.

[Figure]

**Accuracy validation of air temperature measurements and hourly rainfall measurements**

We have added this information in the revised manuscript (Page 25, Line 474-482):

*"Most datasets, including debris-flow kinematic data, particle size distribution of debris flows, rheological data of debris-flow slurry, cross-sectional measurement data, sediment concentration, grain size distribution data, and observation data from runoff plots, were obtained through manual observation. These data are considered to have relatively small errors. Additionally, rainfall data from tipping-bucket rain gauges, collected prior to 2024, showed high reliability when cross-validated with data from regional national meteorological stations. In 2024, piezoelectric rain gauges and meteorological stations were installed and underwent performance verification at national benchmark sites before deployment. The correlation coefficients between these instruments and the reference data from national meteorological stations were 0.951 for the piezoelectric rain gauges and 0.996 for the meteorological stations."* **C6**: There are still several language and formatting errors throughout the manuscript. For instance, in the caption of Figure 1, (c) is incorrectly labeled as (b).

**R6**: We sincerely thank the reviewer for highlighting the language and formatting errors. The error in the caption of Figure 1 has been corrected. Additionally, we have thoroughly proofread the entire manuscript to identify and address any other language and formatting issues.

**C7**: Additionally, the sentence on Line 104 "This paper summarizes of the debris-flow observation at Jiangjia Ravine and overviews of the core data", contains grammatical errors. The authors are advised to carefully proofread the entire manuscript to correct such issues and ensure consistency in figure captions, labeling, and overall language quality.

**R7**: Thank you for highlighting the grammatical errors in our manuscript. We have revised it: "This paper summarizes the debris-flow observations at Jiangjia Ravine and provides an overview of the core data." We have thoroughly proofread the entire manuscript to correct similar grammatical issues. Additionally, we have ensured consistency in the figure captions, labeling, and overall language to enhance the clarity and professionalism of our paper.